cognition/neuroscience/biomechanics

gait, cognitive load, environment, visual discomfort, image statistics

**Author for correspondence:**
D. Burtan
e-mail: daria.burtan@bristol.ac.uk

# The nature effect in motion: visual exposure to environmental scenes impacts cognitive load and human gait kinematics

D. Burtan[1], K. Joyce[1], J. F. Burn[2], T. C. Handy[3], S. Ho[3] and U. Leonards[1]

[1]School of Psychological Science, and [2]Queen's School of Engineering, University of Bristol, Bristol, UK
[3]Department of Psychology, University of British Columbia, Vancouver, British Columbia, Canada

DB, 0000-0002-2881-9005; KJ, 0000-0002-5539-7178; UL, 0000-0001-6143-7466

Prolonged exposure to urban environments requires higher cognitive processing resources than exposure to nature environments, even if only visual cues are available. Here, we explored the moment-to-moment impact of environment type on visual cognitive processing load, measuring gait kinematics and reaction times. In Experiment 1, participants ($n = 20$) walked toward nature and urban images projected in front of them, one image per walk, and rated each image for visual discomfort. Gait speed and step length decreased for exposure to urban as compared with nature scenes in line with gait changes observed during verbal cognitive load tasks. We teased apart factors that might contribute to cognitive load: image statistics and visual discomfort. Gait changes correlated with subjective ratings of visual discomfort and their interaction with the environment but not with low-level image statistics. In Experiment 2, participants ($n = 45$) performed a classic shape discrimination task with the same environmental scenes serving as task-irrelevant distractors. Shape discrimination was slower when urban scenes were presented, suggesting that it is harder to disengage attention from urban than from nature scenes. This provides converging evidence that increased cognitive demands posed by exposure to urban scenes can be measured with gait kinematics and reaction times even for short exposure times.

# 1. Introduction

A wide range of environmental factors such as microclimate, terrain, air and noise pollution have been shown to directly affect individuals' physical and mental well-being (for reviews see [1,2]). Another factor often raised is the importance of exposure to nature or blue-green infrastructure in the form of parks and closeness to water for restorative purposes, with supporting evidence coming from urban planning, human geography, medicine and psychology (e.g. [3–6]). In this context, psychological restoration refers to the ability to replenish one's cognitive resources [7] that deplete by everyday activities in cognitively demanding environments and reduces stress levels [8]. For example, living close to green spaces has been shown to reduce stress and to improve physical and mental well-being alike [9]. Similarly, active commuting through nature environments as opposed to build environments is associated with better mental health [10] and increased cognitive performance [11]. Yet, what exactly it is in such green spaces or in nature more generally that leads to restorative effects and how much exposure is needed to obtain restorative benefits remains unclear. Though some authors have suggested that even factors such as the biodiversity of a green space might play a crucial role (e.g. [12,13]).

Evolutionary theory, in particular the so-called biophilia hypothesis [14], proposes that human beings have an innate need to connect with nature and to affiliate with different forms of biological life. Other psychology-based theories provide more cognitive explanations, such as stress recovery theory (SRT; [8]) or attention restoration theory (ART; [15–17]). SRT puts forward the idea that nature reduces the level of psychological stress, while ART proposes that nature as opposed to urban environments helps to replenish attentional resources. Indeed, there is ample evidence that participants perform better in demanding cognitive tasks after walking in nature for 30 min than after walking for the same time in an urban environment [15,16,18,19].

That at least some of the beneficial effects of exposure to nature are not simply driven by cleaner air or reduced auditory noise but by the visual aspects of the environment is suggested by findings that even exposure to images of nature scenes leads to a measurable improvement in post-exposure attention and memory performance compared with exposure to images of urban scenes [20,21]. This then leads to the central issue in our study: if cognitive benefits can be observed *after* sustained exposure to nature versus urban scenes [20,21], theoretically, such effects should also be observable in real-time measures. Here, we addressed this prediction using measures of gait kinematics.

In particular, evidence from dual-task experiments suggests that increases in cognitive load lead to decreased walking speed in addition to increased variability in stride length and stride timing (for review see [22]). Indeed, dual tasking (e.g. walk and talk) reveals cognitive-motor interference not only in older adults and clinical populations (for review see [22,23]) but also in young healthy adults [24,25]. The higher the neural overlap of the networks involved in cognitive and motor tasks, the stronger the interferences tend to be, an effect that further increases with age (see [26]). The link between perceptual load and gait has been little investigated so far (but see [24,27–29]), but the high reliance of walking on sensory (in particular, visual) input should be particularly sensitive to environmentally induced cognitive load.

Crucially, the potential benefit of this study goes beyond just testing this real-time prediction. As well, it gives us a novel approach for studying and understanding the core factors driving the 'nature' benefit, including basic image statistics such as contrast distributions or fractal content (e.g. [30,31]), differences in attentional demands (e.g. [17,32]), higher visual cognitive aspects such as the meaning of scenes (e.g. [33]), or more general stress [8].

Here, we investigated this idea with two experimental approaches: (i) we measured gait changes for a laboratory-based walking task with trial-by-trial changes in exposure to different visual scenes to obtain an objective measure that can later be used in more real-world environments to measure environmentally induced cognitive load, and (ii) we measured reaction time differences for a basic shape discrimination task in which nature and urban images were introduced as task-irrelevant distractors, allowing us to tap into differences in their ability to capture visual attention.

We also investigated whether basic image statistics and their related visual discomfort ratings contribute to environmentally induced cognitive load. Indeed, nature and urban scenes differ not only with regard to the demands they might pose on higher visual cognition factors such as visual attention but also with regard to their low-level image statistics such as contrast distributions (amplitude of contrast at all visual frequencies) or fractal content (self-repetitive patterns). Contrast distributions have been found to typically fall close to $1/f$ in nature: spatial frequency decreases with increasing amplitude of contrast and vice versa [30]. Urban environments, on the other hand, tend to

have repetitive high-contrast patterns that are far removed from 1/f contrast distributions; for example, high-frequency stripes in paving or crosshatch patterns in brickwork are very common [34]. It has therefore been suggested that the human visual system has evolved to process scenes close to a 1/f distribution with high efficiency [35]. This theory has been supported by research that found images diverging further from a 1/f contrast distribution elicit increased cortical responses and are more likely to induce visual discomfort (e.g. [36–40]). Visual discomfort is an adverse effect when viewing certain visual stimuli that can evoke neurological symptoms such as headache (e.g. [41]), nausea, drowsiness and in extreme cases, pattern-sensitive epilepsy (e.g. [41–43]). We therefore included an analysis of image statistics to see whether differences between the contrast distributions of nature and urban scenes could explain any of the differences found in gait kinematics. Fractals contain internal self-repetitive visual information, which is common in nature environments (e.g. [44]) but seems to be far less common in built environments (e.g. [24]). Higher internal self-repetitive visual information [45] is thought to increase perceptual fluency and to require fewer cognitive resources [24,31]. Any restorative/stress-reducing effect of interacting with nature could thus be related to more efficient sensory processing or sparse coding (e.g. [30,35,46]) due to the number of fractals in the visual scene. Accordingly, the fractal content of images might explain the variability of gait kinematics, if found between nature and urban scenes (see also [24]).

# 2. Experiment 1: measuring changes in gait kinematics to quantify cognitive load differences between nature and urban scenes

The aim of this experiment was to establish the impact of environment type (urban versus nature) on cognitive processing load, using changes in gait kinematics to quantify changes in cognitive load.

## 2.1. Material and methods

### 2.1.1. Participants

Sample size calculations took into account the substantial amount of repetitions within individual participants for all conditions of interest and were based on modelling estimates for within-participant repeated measures correlations provided by Bakdash & Marusich [47]: to obtain 80% power for a medium effect size (0.3) and within-participant repeated paired measures of 20 or more, we needed a minimum of 12 participants. Twenty participants (6 male; aged 18–36 years, $M = 23$ years) took part in this study in the Bristol Vision Institute (BVI) movement laboratory at the University of Bristol. All participants reported normal or corrected-to-normal visual acuity, no injuries or conditions that might have impacted their walking, and all gave their informed written consent prior to commencing the study. The experiment was approved by the Faculty of Science Ethics Committee at the University of Bristol (ref: 27041635961). Participation took place by reimbursement to account for participants' time.

### 2.1.2. Stimuli

For this study, we selected 100 images of nature and urban scenes out of a far larger image set taken by two of the authors, in addition to five plain grey images. Images presented a range of landscapes and urban spaces across Europe and Canada (see experiment 2 in [24], for the same image set). Image resolution was $1280 \times 800$ pixels. First, nature and urban scenes were visually matched as closely as possible for their spatial composition: for this, half of the nature scenes and half of the urban scenes included a walkable path while the other half did not; thus, we had four image categories: nature path (25 images), nature no path (25 images), urban path (25 images) and urban no path (25 images). Secondly, images across all four categories were controlled for perceived depth (distance to the centremost point), perceptually grouping each image category by distance into five image groups: very close, close, medium, far and very far, with five images per distance. Each image within this five by five design had an image in the other four categories that was perceptually matched as closely as possible in its overall spatial layout as agreed on by three of the authors.

For each image, contrast distributions were calculated by applying a model developed by Penacchio & Wilkins [30]. We decided to use this method as it has been used before to measure the link between contrast distribution and visual discomfort. This model calculates the amplitude of

contrast at all visual frequencies (limited by pixels) for all orientations and outputs the residuals after comparison with a typical $1/f$ distribution. Higher residuals reveal a contrast distribution further from $1/f$. For this, images were transformed into greyscale, and then cropped to $800 \times 800$ pixel square to be able to apply the procedure. To obtain a measure for the entire image ($1280 \times 800$ pixels), residuals for the left and right part of the image were calculated separately, and the average residual of the two image parts was taken as the value for this stimulus' residual (note that as total image size is $1280 \times 800$ pixels, there is a substantial spatial overlap between the two image halves used to calculate contrast). In line with earlier findings [30], urban images had significantly higher residuals ($M = 2.4 \times 10^{14}$, s.d. $= 9.4 \times 10^{13}$) and thus sat further away from a $1/f$ distribution than nature images ($M = 1.8 \times 10^{14}$, s.d. $= 9.0 \times 10^{13}$); ($t_{98} = 3.241$, $p < 0.01$, $d = 0.65$).

Fractal dimensions of our images were taken from the calculations described in Ho *et al.* [24] for the same image set (see their experiment 2). In brief, calculations were based on the Minkowski–Bouligand fractal dimension box-counting technique [48]: after normalizing colour images and converting them into greyscale images, images were binarized using the mean image value before running a box-counting algorithm over a range of box sizes to calculate fractal dimensions.

### 2.1.3. Procedure

On arrival, participants were given written and verbal explanations of the experiment, before having small spherical retro-reflective markers attached to their shoulders (lateral clavicle), knees (patella), outside of their ankles (lateral malleolus) and their feet (first metatarsophalangeal joint). In addition, participants were given an elasticated belt to wear at hip height with three markers to locate the left hip, right hip and lower abdomen (hereon referred to as hip). The location of these markers was detected by a motion capture system (Oqus, Qualisys AB, Sweden) with a recording frequency of 100 Hz. The system consisted of 12 cameras and was calibrated prior to testing each participant, leading to a typical spatial accuracy of 1 mm$^3$ across a captured space of $12 \times 2 \times 2.4$ m. The room was dimly lit throughout the experimental session with blackout curtains all along the long sides of the room.

Following this preparation procedure, participants took part in the experiment. Note that the experiment was divided into two parts: (i) a cognitive-motor interference task with walking as the motor task and verbal trail making as the cognitive task—this part served as a control to establish whether our methodology was sufficiently robust to observe changes in gait kinematics associated with changes in cognitive load [22], and (ii) the environment-induced perceptual load motor interference task as the task of interest. The order of the two experimental parts was counterbalanced across participants. The procedure and the results of the cognitive-motor interference task are presented in the electronic supplementary material.

*Environmentally induced perceptual load—motor interference task*
Participants were required to walk repeatedly down the 15 m laboratory at their natural walking speed, starting to walk from a cross marked on the floor at one end of the laboratory immediately after the appearance of an image on the wall at the opposite side of the laboratory. For each walk, one of the following images was displayed: a nature scene (50), an urban scene (50) or a neutral grey image (5). The image display size was 3 m wide $\times$ 2 m high, corresponding to $11.4° \times 7.6°$ of visual angle when viewed from the starting point of the walk, and $57° \times 38°$ of visual angle when viewed from the end line of the three-dimensional motion capture space.

After each walk, participants were asked to rate the image seen for its visual discomfort. Visual discomfort ratings were given on a 7-point Likert scale from '1—extremely comfortable to view' to '7—extremely uncomfortable to view' (participants had been familiarized with this scale and the definition of visual discomfort during the verbal briefing).

After rating the image, the participant returned to the starting cross-location for the next trial. All 105 stimuli (50 nature scenes, 50 urban scenes and 5 grey control images) were presented in a randomized order. The task took approximately 40 min to complete, and participants were offered a break at the halfway point (after trial 52). Participants could ask for additional breaks, if they deemed it necessary.

Gait was recorded for each walking trial, using the motion capture system (x-direction depicting lateral movement, y-direction depicting the direction of travel down the laboratory, z-direction depicting vertical movement).

At the end of the experiment, participants were thanked for their contribution and debriefed.

## 2.2. Data analysis

### 2.2.1. Three-dimensional motion capture data

A pre-processing procedure was applied to the raw motion capture data. Raw data were pre-processed using proprietary software (QTM, Qualisys AB) automatically to identify the trajectories of markers. During normal walking, we expected steps of a typical length alternating between the left and the right foot. If the analysis revealed missing markers, steps over 1.3 m in length or consecutive steps from the same foot, this was highlighted as inconsistency. Such trials were manually checked, and errors in the labelling of the markers by the model were corrected where appropriate and possible. Any trials with missing sensor data (foot and hip markers) were removed from further analysis. A low-pass filter was applied to the raw data of interest, i.e. the foot and hip markers, to remove high-frequency noise. Specifically, a bidirectional second-order Butterworth filter was applied with a cut-off frequency of 5 Hz. Data were truncated for each trial to remove all data from the first 0.5 m and the last 2 m of captured space (i.e. 5 m before the wall on which images had been presented), leaving 9.5 m for gait analysis per trial. This excluded those parts of each trial where the subject was accelerating or decelerating so that walking speed was approximately constant in the data used for subsequent analysis.

### 2.2.2. Step detection and measures of gait

From the kinematic data derived from each trial, key information was extracted from the velocity and position data of the foot markers to label individual steps. Steps were defined as the stationary periods for each foot; i.e. when the marker moved less than 5 cm in 0.1 s. The position of a step was determined by the position of the foot marker in the middle of this stationary period. The landing time was therefore labelled as the time corresponding to the maximum deceleration of the foot on the Y-axis prior to this stationary period and the lifting time as the point of maximum acceleration on the Y-axis post stationary period.

Subsequently, measures of gait were calculated from the pre-processed data for feet and hip. Specifically, trial velocity was calculated as the distance the hip marker travelled (i.e. 9.5 m) divided by the time taken to complete the walk. Step length was defined as the distance from the foot marker on one foot to the foot marker on the other foot at landing time and was calculated by subtracting the Y-position of the rear foot from the Y-position of the forward-stepping foot. Stride time was defined as the difference between one landing time and the subsequent landing time of the same foot. Finally, swing time was defined as the difference between the lifting time and the corresponding landing time of an individual foot. These data were then summarized as mean velocity, step length, stride time and swing time for each trial. The variability in step length, stride time and swing time for each trial was also calculated. Note that for the first step for each foot as well as the last step, stride time, step length and swing time were undefined due to an indeterminate lifting time/start position of the rear foot or an indeterminate landing time/position of the front foot, respectively (as these were outside the measured area). As such, only the data for steps measured in their entirety were used for analysis.

### 2.2.3. Exclusion criteria

One participant's data were excluded from analysis as they did not properly follow the instructions for the task. One further participant was excluded from analysis due to having an unusual walking style (mean step length >2.5 s.d. from the group mean) affecting too many trials. This left 18 participants' datasets for analysis; (5 male), aged 18–34 ($M = 22$).

For these 18 participants, individual trials were excluded on the basis of missing data (unlabelled markers) or if a participant accidentally stopped their walk before reaching the end of the motion capture space. As these errors were only detected during the analysis stage, such trials could not be repeated. On rare occasions, synchronization between stimulus computer and motion capture system failed. In these situations, the affected trial was repeated. The inclusion criterion was a minimum of 43 trials per environment condition (nature/urban) and 4 trials per control condition. The mean number of trials per nature environment per participant for step analysis was 49.28 (s.d. = 1.87) and for velocity analysis 49.44 (s.d. = 1.69) out of 50. The mean number of trials per urban environment per participant for step analysis was 49.44 (s.d. = 1.46) and for velocity analysis 49.50 (s.d. = 1.47) out of 50. The mean number of trials per neutral condition per participant for both step and velocity

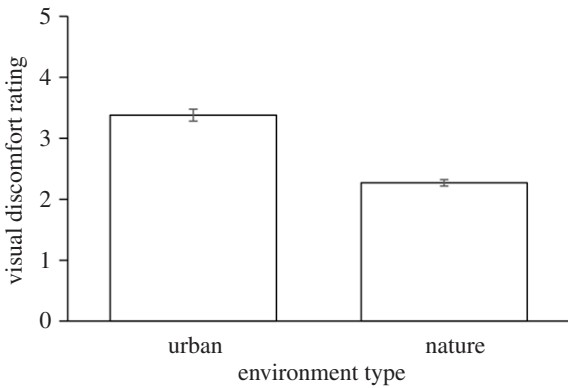

**Figure 1.** Group averages of individual visual discomfort ratings per image (7-point Likert scale) for the two environment types: nature and urban. Error bars reflect ±1 s.e.m.

analysis was 4.89 (s.d. = 0.32) out of 5. In addition, for the multi-level modelling analysis outliers were removed from the data (greater than 2.5 s.d. from the group mean).

Image set and data are available at the University of Bristol's data repository, data.bris, at https://doi.org/10.5523/bris.36buvm31514jl2o7inlv4dhtod.

## 2.3. Results

### 2.3.1. Environmentally induced perceptual load—motor interference task

#### Visual discomfort ratings

In line with expectations from earlier studies on visual discomfort [34], subjective visual discomfort ratings were higher for urban images ($M = 3.38$, s.d. = 0.7) than for nature images ($M = 2.27$, s.d. = 0.38), see figure 1. A one-way ANOVA with repeated measures revealed that this effect was highly significant ($F_{1,49} = 119.583$, $p < 0.001$).

#### Gait data

Repeated measures MANOVAs were applied to the gait data of this part of the experiment, adding an order of experimental parts as a between-subjects variable and environmental stimulus type (Urban/Nature/Neutral) as a within subjects variable for seven dependent gait measures (mean velocity, mean step length, mean stride time, mean swing time, variability of step length, variability of stride time and variability of swing time).

Velocity: Analysis revealed a significant main effect of environment on mean velocity ($F_{2,32} = 32.34$; m.s.e. < 0.05; $p < 0.001$, $\eta_p^2 = 0.67$), see figure 2a. *Post hoc* tests with Bonferroni correction revealed a significant difference between all three environmental conditions; participants walked fastest toward neutral images, significantly slower toward nature images ($p < 0.01$) and significantly slower again toward urban images (neutral–urban: $p < 0.001$, nature–urban: $p < 0.05$).

Step length: Analysis with Greenhouse–Geisser correction showed that there was also a significant main effect of environment on mean step length ($F_{1.47,23.58} = 23.55$; m.s.e. < 0.05; $p < 0.001$, $\eta_p^2 = 0.60$), see figure 2b. *Post hoc* tests with Bonferroni correction revealed significant differences between all conditions; neutral images resulted in the longest mean step length, with a significantly shorter step length for nature images ($p < 0.01$) and significantly shorter step length again for urban images (neutral–urban; $p < 0.001$, nature–urban; $p < 0.01$).

Stride time: In addition, analysis with Greenhouse–Geisser correction revealed that there was a significant main effect of environment on mean stride time ($F_{1.39,22.30} = 29.33$; m.s.e. < 0.01; $p < 0.001$, $\eta_p^2 = 0.65$), see figure 2c. *Post hoc* tests with Bonferroni correction showed that neutral images elicited shorter stride times than both nature ($p < 0.01$) and urban images ($p < 0.001$). However, there was no significant difference in stride time for nature and urban images ($p > 0.05$), suggesting that this measure is less sensitive than overall velocity and step length.

Swing time: In addition, analysis with Greenhouse–Geisser correction revealed that there was a significant main effect of environment on mean swing time ($F_{1.25,20.04} = 7.66$; m.s.e. < 0.001; $p < 0.01$, $\eta_p^2 = 0.32$), see figure 2d. *Post hoc* tests with Bonferroni correction showed that neutral images elicited

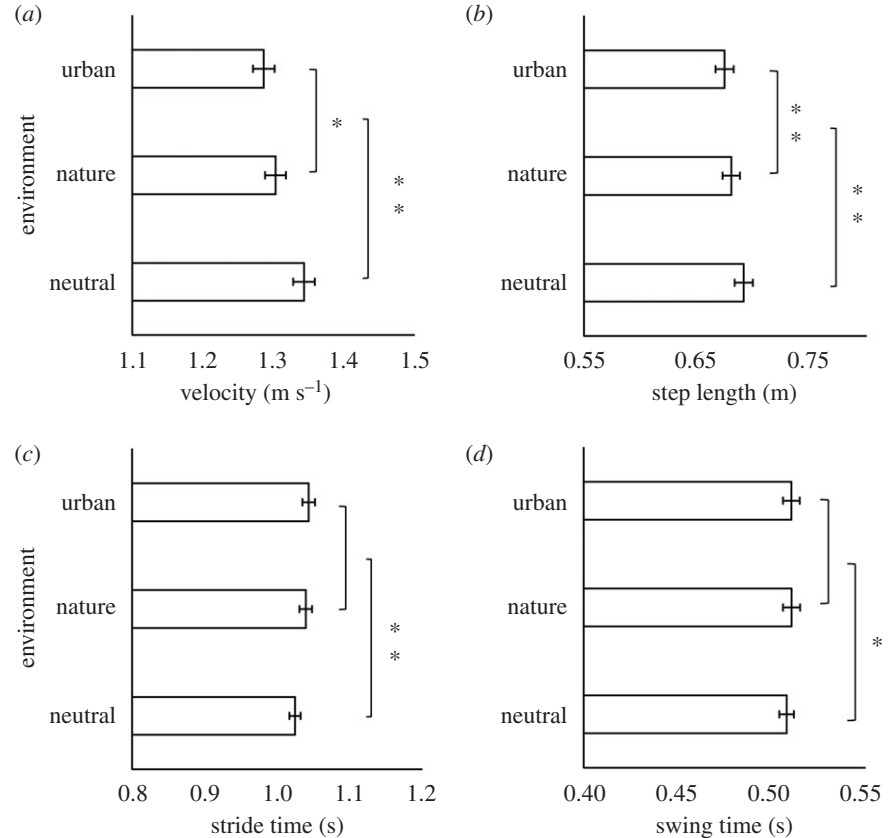

**Figure 2.** Group averages of individual mean (*a*) velocity (metres per second), (*b*) step length (in metres) (*c*) stride time (in seconds) and (*d*) swing time (in seconds) across environment type. Error bars reflect ±1 s.e.m. $^*p < 0.05$, $^{**}p < 0.01$.

shorter swing times than both nature ($p < 0.05$) and urban images ($p < 0.05$). However, there was no significant difference in swing time for nature and urban images ($p > 0.05$), suggesting that this measure is also less sensitive than velocity and step length.

There was no main effect of environment on the variability of step length, the variability of stride time and the variability of swing time.

Experimental part order did not affect any of the gait measures. However, there was a significant interaction between experimental part order and environment type for velocity ($F_{2,32} = 5.01$; m.s.e. < 0.01; $p < 0.05$, $\eta_p^2 = 0.24$) and stride time ($F_{1.39,22.30} = 4.57$, m.s.e. < 0.001; $p < 0.05$, $\eta_p^2 = 0.22$) (see electronic supplementary material for more detail).

### 2.3.2. Multi-level modelling

#### Velocity

To tease apart possible effects of image statistics (i.e. contrast distributions and fractal content), subjective visual discomfort, image configuration (presence of a walkable pathway in the image) and environment type on walking speed, we applied a cross-classified multi-level model to the velocity data. As a composite measure of both step length and stride time, we reasoned that velocity might be the most sensitive gait measure to assess environmentally induced cognitive load.

All continuous data were transformed into Z-scores for ease of interpretation, and categorical variables (pathway and environment) were dummy coded to reflect the change from a reference category (path and nature, respectively).

Exclusions: neutral image trials were excluded from this analysis due to missing data for all predictors (i.e. image statistics, image configuration and subjective discomfort ratings) other than environment. In addition, 18 trials were excluded due to outlier screening (see Material and methods section).

A series of models was fitted through four stages to establish the model of best fit. After each stage, the significance of each fixed effect (predictor) was assessed with chi-squared statistics, and insignificant predictors were discarded.

**Table 1.** Model fit comparisons for models estimating velocity from the characteristics of the image viewed. Note. PT, participant; IM, image; ENV, environment; DIS, discomfort rating; IMS, image statistics (1/f residuals); PA, path; FD, fractal dimension; DIC, deviance information criterion. Model lettered 'a' show the best combination of predictors at each stage after discarding insignificant predictors.

| model | DIC | fixed | random |
| --- | --- | --- | --- |
| 1 | 3483.390 | | PT, IM |
| 2 | 3422.810 | ENV, DIS, IMS, FD, PA | PT, IM |
| 2a | 3420.733 | ENV, DIS, IMS | PT, IM |
| 3 | 3419.824 | ENV, DIS, IMS, ENV × DIS, ENV × IMS, DIS × IMS | PT, IM |
| 3a | 3422.476 | ENV, DIS | PT, IM |
| 4 | 3419.996 | ENV, DIS, ENV × DIS | PT, IM |
| 4a | 3419.996 | ENV, DIS, ENV × DIS (Model 4) | PT, IM |

Model 1: a cross-classified model was created with participant and image crossed at level two as random effects and trial at level one as a random effect, i.e. each individual trial was treated as a case ($N = 1764$; on average $98 \pm 3.16$ s.d. trials per participant).

Model 2: predictors (environment, discomfort, image statistics: 1/f residuals, fractal dimensions, path) were added as fixed effects.

Model 3: all relevant two-way interactions were added as fixed effects.

See table 1 for all models fitted; models lettered 'a' show the best combination of predictors at each stage, following the discarding of insignificant predictors.

The model of best fit was selected from the final lettered 'a' models based on deviance information criterion (DIC) statistics. A lower DIC equates to a better fit. The best-fitting model was therefore 4a, parameter estimates for this model are displayed in table 2.

Environment ($\chi^2_1 = 2.619$, $p < 0.01$), visual discomfort ($\chi^2_1 = 36.604$, $p < 0.001$) and interaction between environment and visual discomfort ($\chi^2_1 = 4.276$, $p < 0.05$) were significant predictors of velocity, with people walking more slowly when walking towards images of urban environments and images they perceived as more uncomfortable to look at. Subjective discomfort and environment seem heavily interrelated and explained some of the variability in walking speed.

Basic image statistics (contrast distribution and fractal dimension) and presence of walkable path did not improve model fit (higher DIC statistics). These factors did not explain any of the variance over and above the other predictors.

## 2.4. Discussion experiment 1

This experiment provided the first evidence that exposure to visual scenes of urban environments as compared with nature environments requires higher amounts of cognitive/perceptual resources: indeed, participants walked more slowly and with smaller steps toward urban scenes as compared with nature scenes, mirroring behaviour usually described for dual-task conditions in which walking is affected by the second task requiring higher cognitive resources ([22–24]; also see the control experiment in the electronic supplementary material). The differences in cognitive/perceptual load requirements evoked by the two types of visual environment are thus big enough to be picked up on a trial-by-trial basis, using changes in gait kinematics as an objective measure.

While these results support our main hypothesis, what might contribute to the increase in cognitive load requirements for exposure to urban scenes was less clear: basic image statistics, visual discomfort or image type.

Multi-level modelling revealed that environment, visual discomfort and the interaction between environment and visual discomfort explained some of the variances in gait kinematics (i.e. walking speed). Neither image contrast distributions nor fractal dimensions had a predictive value on gait kinematics, speaking against differences in basic image statistics between nature and urban scenes being at the core of nature's benefits or higher cognitive demands for urban images.

However, interpretations of experimental outcomes need to be treated with caution: subjective visual discomfort was significantly higher for urban scenes than for nature scenes. This makes it impossible to

**Table 2.** Fixed effects estimates (top) and random effect variance estimates (bottom) for the model with best fit (Model 4a; table 1) predicting velocity from the characteristics of the image viewed. Note. Estimates reflect size of the effect on standardized velocity. Burn-in = 500, chain length = 10 000. Degree of freedom is 1 for all Chi-square $(\chi_1^2)$ statistics. $^{***}p < 0.001$, $^{**}p < 0.01$, $^*p < 0.05$.

| | | | 95% CI | | |
| parameter | estimate | s.e. | lower | upper | $\chi_1^2$ |
| --- | --- | --- | --- | --- | --- |
| *fixed* | | | | | |
| intercept | 0.314 | 0.190 | −0.097 | 0.653 | 1.648 |
| environment (urban) | −0.222 | 0.085 | −0.388 | −0.056 | 2.619** |
| discomfort | −0.174 | 0.029 | −0.231 | −0.118 | 36.604*** |
| DIS × ENV | 0.076 | 0.037 | 0.004 | 0.148 | 4.276* |
| *random* | | | | | |
| participant | 0.637 | 0.252 | 0.315 | 1.261 | |
| image | 0.004 | 0.003 | 0.001 | 0.010 | |
| walk | 0.399 | 0.014 | 0.373 | 0.427 | |
| deviance information criterion (DIC) | | | | | 3419.996 |

establish just how much of the changes in gait kinematics were due to environment type *per se* and how much was due to our discomfort rating task. Indeed, participants might have found it more difficult to rate urban images for visual discomfort than nature images, slowing their gait while trying to perform this rating; albeit we asked participants to rate images for their visual discomfort only after they had finished their walk, we cannot exclude that they started to rate images already while they were still walking. Such an interpretation is fully in line with discomfort ratings by Ho and co-workers for the same image set: participants' reaction times were longer when rating urban images for visual discomfort than rating nature images (see experiment 2 in [24]). It thus remains an open question whether simple exposure to different image types without discomfort ratings would also be sufficient to affect gait kinematics on a trial-by-trial basis.

Our data suggest that not basic image processing differences related to low-level image statistics such as contrast distributions [30] or fractal dimensions [24], but rather higher visual cognition processes are at the core of differences in cognitive load induced by nature and urban images. However, we cannot exclude that the range of fractal dimensions and contrast distributions tested was simply not big enough to be picked up with our current set of stimuli. Also, our results do not mean that other low-level image properties such as saturation, hue or luminance contrast might not contribute to changes in cognitive load and gait. These questions will have to be explored further in future experiments.

# 3. Experiment 2: exploring attentional capture for urban and nature images

Outcomes of Experiment 1 were in line with our hypothesis that urban images pose higher demands on cognitive processing than nature images due to higher level visual stimulus associations, an effect that was pronounced enough to be measured as changes in gait kinematics on a trial-by-trial basis.

To investigate whether these differences in processing demands could be explained by an increased ability of urban scenes to capture attention, we ran a second experiment in which participants were asked to perform a simple visual shape discrimination task in the presence of the same images of nature or urban environments used in Experiment 1; this time as task-irrelevant distractors (see figure 3). We reasoned that if the scene content of urban environments were indeed to capture people's attention more readily than the scene content of nature environments, this would require higher amounts of processing power to disengage from them and perform the task at hand, thus slowing participants' responses in the unrelated shape discrimination task (see also [21]).

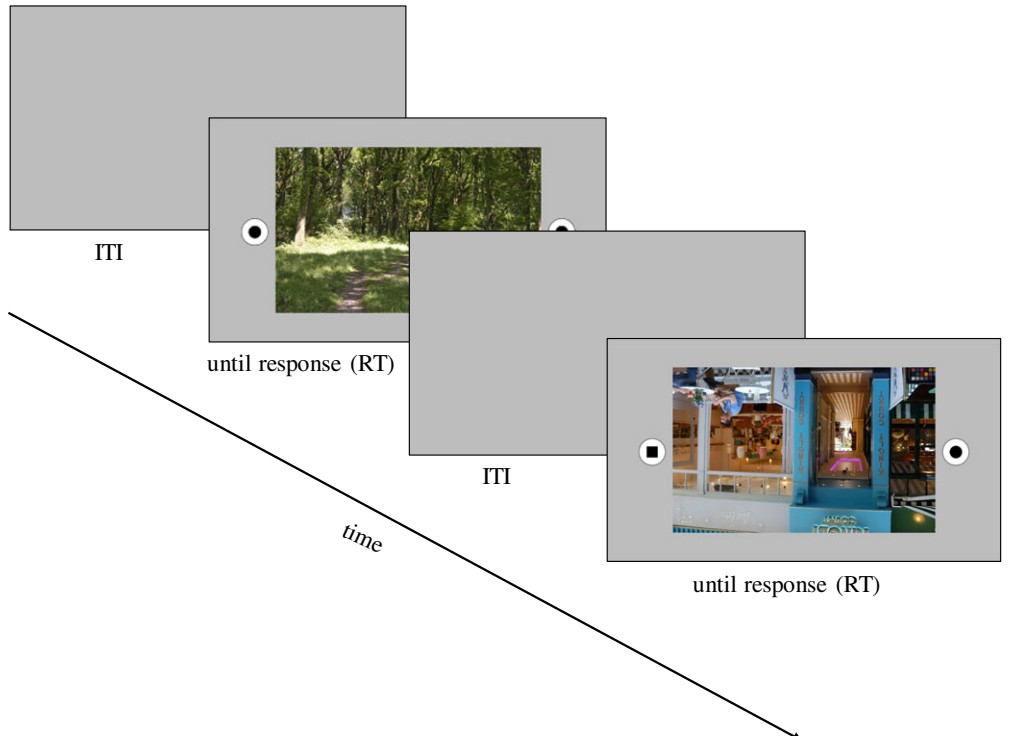

**Figure 3.** Shape discrimination task. The presentation of a central fixation cross for a random duration: ITI, inter-trial interval 0.7, 0.8, 0.9, 1.0, 1.1, 1.2, 1.3 (in seconds); RT, reaction time (in seconds).

## 3.1. Methods and materials

### 3.1.1. Participants

Sample size calculations were based on two different assumptions to allow different analyses: (i) taking our within-participant repeated measures design with multiple repetitions per condition into account as required for multi-level modelling, we based the first calculation on correlation modelling outcomes provided by Bakdash & Marusich [47]: to obtain 80% power for a medium effect size (0.3) and a number of intra-individual repeated paired measures of 20 or more, we would need a minimum of 12 participants; (ii) for a more classical repeated measures ANOVA based on mean values per condition per participant, a similar power calculation (0.8 power) for a medium effect size of 0.3 and a conservative assumption of repeated measures correlation of 0.5 would require 24 participants for individual main effects as well as the interaction effect. Forty-five participants (8 males aged between 18–29 years with a mean age of 21 years, and 37 females aged between 16 and 54 years with a mean age of 22 years) took part in this study at the University of Bristol. All participants reported normal or corrected-to-normal visual acuity and normal colour vision. All gave written informed consent at the beginning of the study. Participants took part in the experiment for course credit. The experiment was approved by the Faculty of Life Sciences' Ethics Committee at the University of Bristol (ref. 28071871142).

### 3.1.2. Material and task

The task consisted of a basic shape discrimination task (figure 3).

Participants were asked to decide as quickly and as accurately as possible whether two geometric shapes presented to the left and right of a task-irrelevant photographic image presented in the centre of the screen were identical or different, by pressing the according key on a keyboard in front of them. The two shapes were a black circle (diameter of 61 pixels) and a black square (54 × 54 pixels), i.e. matched in their overall number of pixels. Each shape was presented in the middle of a white circle (diameter of 130 pixels). The task-irrelevant photographic images in the centre of the screen consisted of the same 50 nature and 50 urban scene images used in Experiment 1.

The photographic images subtended an area of 34° × 21° of visual angle. Shapes within their white circles subtended 3.5° of visual angle and were presented 20° degrees of visual angle from the centre

of the screen, i.e. the outer line of the circle was 1.253° degrees of visual angle away from the corners of the photographic image. There were four shape-pair conditions: circle (L)–circle (R), square (L)–square (R), circle (L)–square (R), and square (L)–circle (R); each of which was presented equally often, but in random order. The screen background was a medium grey of average luminance (94.24 cd m$^{-2}$) and subtended an area of $51° \times 29°$ of visual angle.

To distinguish between the impact of image statistics and associated higher level cognitive image associations on task-unrelated distraction, each image was presented once upright and once in an inverted orientation, resulting in a total of 200 trials. Image statistics remain the same irrespective of stimulus orientation while we reasoned that it should be more difficult to detect automatically the gist of a scene when upside down and thus reduce its ability to capture attention. Any reaction time differences for our shape discrimination task in the presence of upright as compared with inverted images should thus be due to cognitive demands associated with image meaning.

### 3.1.3. Procedure

On arrival, participants were given written and verbal information about the study and were then seated in front of the computer on which the experiment was run, with a viewing distance of 57 cm to the 21-inch monitor. The testing room was quiet with dimmed lighting. After participants had adapted to the background luminance levels of the room, the actual experiment began.

Each trial started with the presentation of a central fixation cross for a random duration of between 0.7 and 1.3 s. This was followed by the presentation of one of the 200 photographic images (50 nature upright, 50 nature inverted, 50 urban upright and 50 urban inverted) centred between the two shapes for the shape discrimination task; which of the four shape combinations and which photographic image were presented, was pseudo-randomly determined. Images stayed on the screen until participants responded by pressing the according key on the keyboard. If participants pressed the wrong key, a short beep alerted them of their mistake.

There was one break during the study halfway through, i.e. after 100 trials. Response accuracy and reaction times were recorded.

### 3.1.4. Data analysis

The first trial for each participant was removed as a practice trial. Participants with a task accuracy below 80% were excluded from the analysis. This left 41 datasets for analysis (7 males aged between 19–29 years with a mean age of 21 years, and 34 females aged between 16 and 54 years with a mean age of 23 years; group mean age of 22). Per participant, the median reaction times for the four stimulus distractor conditions (nature upright, nature inverted, urban upright, urban inverted) were calculated from 5% trimmed data.

## 3.2. Results

A 2 (environment) $\times$ 2 (image orientation) repeated measures ANOVA with median reaction times as dependent variable revealed a significant main effect of environment (nature versus urban) ($F_{1,40} = 23.111$, $p < 0.05$, $\eta_p^2 = 0.366$): participants performed the shape discrimination task significantly slower during exposure to urban images ($M = 0.79$ s, s.d. $= 0.14$) as compared with nature images ($M = 0.76$ s, s.d. $= 0.13$). There was no significant main effect of orientation (upright versus inverted) on median reaction times ($F_{1,40} = 1.041$, $p > 0.05$, $\eta_p^2 = 0.025$) nor was there a significant interaction between environment type and orientation ($F_{1,40} = 0.001$, $p > 0.05$, $\eta_p^2 = 0.000$) (figure 4).

### 3.2.1. Multi-level modelling

Finally, a cross-classified multi-level model was applied to the data to tease apart possible effects of environment type, image orientation, contrast distributions and fractal content as predictors of variation in reaction times.

All continuous data were transformed into Z-scores for ease of interpretation, and categorical variables (environment, orientation) were dummy coded to reflect the change from a reference category.

A series of models were fitted through four stages to establish the model of best fit. After each stage, the significance of each fixed effect (predictor) was assessed with chi-squared statistics, and insignificant predictors were discarded.

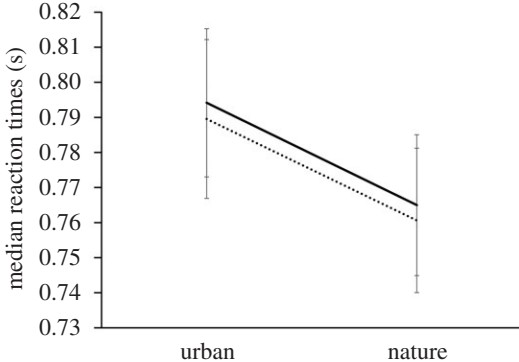

**Figure 4.** Group mean of median reaction times (in seconds) across environment (urban, nature) and orientation type (upright: dotted line; inverted: solid line). Error bars reflect ±1 s.e.m.

**Table 3.** Model fit comparisons for models estimating reaction time from the characteristics of the image viewed. PT, participant; IM, image; ENV, environment; ORI, orientation; IMS, image statistics (i.e. 1/f residuals); FD, fractal dimension; DIC, deviance information criterion. Models lettered 'a' show the best combination of predictors at each stage after discarding insignificant predictors.

| model | DIC | fixed | random |
|---|---|---|---|
| 1 | 18007.577 | | PT, IM |
| 2 | 17998.087 | ENV, ORI, IMS, FD | PT, IM |
| 2a | 17995.037 | ENV, FD | PT, IM |
| 3 | 17996.429 | ENV, FD, ENV × FD | PT, IM |
| 3a | 17999.355 | ENV | PT, IM |

> Model 1: a cross-classified model was created with participant and image crossed at level two as random effects and individual trial at level one ($n = 7346$) as a random effect.
> Model 2: predictors (environment, orientation, image statistics: 1/f residuals, fractal dimension) were added as fixed effects.
> Model 3: all relevant two-way interactions were added as fixed effects.

Table 3 shows the results for all models fitted; models lettered 'a' show the best combination of predictors at each stage, following the discarding of insignificant predictors.

The model of best fit was selected from the lettered 'a' models based on the DIC statistic. A lower DIC equates to a better fit. The best-fitting model was therefore 2a, with the environment (urban) and fractal dimension as significant predictors. Parameter estimates for this model are displayed in table 4.

As expected, environment type was highly predictive of reaction times ($\chi_1^2 = 26.235$, $p < 0.001$). Also, low-level image statistics (i.e. fractal dimensions) predicted reaction times ($\chi_1^2 = 8.427$, $p < 0.01$); yet, predictions went in the opposite direction of what we would have expected: increasing fractal dimensions predicted increasing reaction times.

Including contrast distribution (1/f residuals) as image statistics did not improve the model fit (see increased DIC statistics for Model 2 as compared with Model 2a, meaning that the model that included contrast distributions as image statistics did not explain as much variance as the model that included environment and fractal dimension only.

Due to the unexpected result of increasing reaction times with increasing fractal dimensions in Model 2a when combined with the environment, we wondered whether this result might be due to an unexpected interaction between image type and fractal dimensions. We therefore created two further models to investigate the impact of environment and fractal dimensions separately. The two models were identical in structure to Model 2a but each of them contained only one of the two fixed predictors. The independent estimates for each predictor are outlined in table 5.

These additional analyses revealed that the environment remained predictive of reaction time even in the absence of another predictor. Fractal dimensions, in contrast, did not seem to have predictive power, $p > 0.05$, on their own.

**Table 4.** Fixed effects estimates (top) and random effect variance estimates (bottom) for the model with the best fit (Model 2a; table 4) predicting reaction time from the characteristics of the image viewed. Note. Estimates reflect the size of the effect on standardized reaction times. Burn-in = 500, chain length = 10 000. Degree of freedom is 1 for all Chi-square $(\chi^2_1)$ statistics. ***$p < 0.001$, **$p < 0.01$, *$p < 0.05$.

| parameter | estimate | s.e. | 95% CI | | $\chi^2_1$ |
| --- | --- | --- | --- | --- | --- |
| | | | lower | upper | |
| *fixed* | | | | | |
| intercept | −0.052 | 0.096 | −0.233 | 0.141 | 0.298 |
| environment (urban) | 0.143 | 0.028 | 0.088 | 0.199 | 26.235*** |
| fractal dimension | 0.041 | 0.014 | 0.013 | 0.068 | 8.427** |
| *random* | | | | | |
| participant | 0.347 | 0.084 | 0.219 | 0.546 | |
| image | 0.008 | 0.003 | 0.004 | 0.014 | |
| trial | 0.669 | 0.011 | 0.647 | 0.691 | |
| deviance information criterion (DIC) | | | | | 17995.037 |

**Table 5.** Estimates from independent models for fractal dimension (FD) and environment (ENV).

| estimate | | |
| --- | --- | --- |
| | FD | ENV |
| | −0.000 | |
| | | 0.096*** |

## 3.3. Discussion Experiment 2

In line with the hypothesis that images of urban environments capture attention more readily than nature images and are more difficult to disengage from [21], our results revealed that participants were slower in taking a simple shape discrimination decision when exposed simultaneously to distracting urban images as compared with nature images. Intriguingly, this effect was similarly pronounced for both upright and inverted images, suggesting that at least some low-level basic image statistics might contribute to this effect and not just higher cognitive processes evoked by the meaning of the images.

The results of multi-level modelling revealed that both environment and low-level properties of images (fractal dimension) were predictive of reaction times when included in the same model, thus affecting reaction times. Contrary to expectation, increased fractal dimensions predicted increased rather than decreased reaction times when included in a model with environment type as a main predictor. On their own, however, fractal dimensions did not seem to serve as a reliable predictor for changes in reaction time while environment type did. As for the gait study before, this suggests that there might exist a complex relationship between low- and high-level visual processes involved in the impact of the environment on cognitive processing.

## 4. General discussion

In two experiments using fundamentally different approaches, we presented converging evidence that urban and nature scenes matched for their overall image configuration but selected for content differ in their cognitive processing requirements as suggested by models such as Ulrich's stress recovery theory [8] or Kaplan's attention restoration theory [15,16]. Crucially, we showed that this effect can be measured on a trial-by-trial basis, using changes in gait kinematics or reaction times as measures of cognitive load. In other words, consistent with our theoretical prediction, the cognitive impacts of sustained exposure to nature versus urban scenes [20,21] do in fact present in real time. Given our conclusion and findings, several critical questions and issues follow.

First, in trying to disentangle the respective contributions of low-level and higher-level perceptual processes involved in this effect of image type, our findings were less straightforward and ultimately inconclusive. Environment, visual discomfort and its interaction with the environment explained by far the largest amount of variability in response differences (see Experiment 1). Low-level stimulus characteristics such as fractal content (e.g. [24]) or contrast distributions [30], by contrast, seemed to have little to no explanatory power on behavioural changes; yet, we cannot fully exclude them as (i) our image set had not been chosen with variability in fractal content and contrast distributions in mind, thus potentially not covering sufficient variability, and (ii) there might be a nonlinear interaction between low- and high-level processes, with high-level processes masking more subtle low-level ones. Future studies might therefore vary fractal content/contrast distributions of images parametrically without high-level semantic content to see how such low-level stimulus characteristics impact gait and reaction times in their own rights, in addition to investigating the relationship between low- and high-level perceptual processes in more detail.

A second point is that the results of Experiment 1 make it tempting to speculate on the exact link between visual discomfort and cognitive load changes induced by environment type. It might simply be more difficult to rate urban images for subjective discomfort, thus slowing people's decision making and their gait. While such an explanation cannot be excluded for our gait data, the data from Experiment 2, i.e. the simple shape discrimination task, would indicate that differences are image type-specific rather than related to the task. An alternative explanation for gait slowing and decreased step length might be that viewing images that are more uncomfortable to look at lead to perceptual distortions and other physiologically unpleasant/aversive symptoms which, in turn, make it more difficult to approach such a stimulus. In other words, stimulus aversion rather than attentional capture might be at the core of gait slowing as well as explain the differences seen between urban and nature images in our simple shape discrimination task. In such a scenario, it wouldn't be the restorative effects of nature (i.e. stress reduction) and the corresponding positive emotions through interaction with nature environments, as suggested by Ulrich's stress recovery theory, SRT [8], but a stress factor in urban scenes that explains the differences in urban and nature processing.

However, SRT assumes that affective and aesthetic reactions to environments are not isolated processes, but closely linked to each other as well as to cognitive abilities, emotions, actions and physiological activity [8]. This, in turn, allows for yet another interpretation of our data: not only did participants find urban images more uncomfortable to look at (see Experiment 1 for discomfort ratings), but it is very likely that they also found our urban images less aesthetically pleasing than the nature images we presented (see [49]). If so, environment type would be perfectly confounded with subjective discomfort and/or aesthetics—not only in our study but also in any other study of the psychological benefits of exposure to nature (see [21]). To distinguish between discomfort, aesthetics and environment type *per se*, future experiments should aim to match their stimulus material for aesthetics or discomfort ratings. As images of different environments have a lot of information content, using mobile eye-tracking in future studies would allow investigation of what is capturing participants' attention while they walk towards images of nature and urban scenes.

In conclusion, the two studies reported here present compelling evidence that nature and urban image material pose different demands on cognitive processing that can be quantified on a moment-to-moment basis, using reaction times or gait kinematics as measures. This opens a new line of research to further our understanding of the causal mechanisms underlying health benefits through exposure to nature. What exactly is it in green spaces or nature more generally that promotes restorative effects on cognition, and how much exposure is needed to obtain such effects? With the ability to use changes in gait kinematics as a promising way to objectively quantify and track environmentally induced cognitive load, our results point toward a powerful new approach for addressing these important questions at the nexus of research in cognition and environmental design.

Ethics. Both experiments were approved by the Faculty of Life Sciences' Ethics Committee at the University of Bristol (refs: 27041635961 and 28071871142).

Data accessibility. Image set and data are available at the University of Bristol data repository, data.bris, at https://doi.org/10.5523/bris.36buvm31514jl2o7inlv4dhtod.

Authors' contributions. U.L. and T.C.H. had the original idea for this study, collected the stimulus material and developed the study concept together with K.J. All authors contributed to the final study design. Testing and data collection were performed by K.J. and D.B. K.J. and D.B. performed the data analysis and interpretation under the supervision of U.L. and J.F.B. with additional input by T.C.H. and S.H. D.B. drafted the manuscript, and all other authors provided critical revisions. All authors approved the final version of the manuscript for submission.

Competing interests. We declare we have no competing interests.

Funding. Work in the BVI movement laboratory was supported by the Wellcome Trust (WT089367AIA). D.B. is supported by a PhD studentship from the Faculty of Science, University of Bristol.

Acknowledgements. The authors would like to thank Dr David Redmill for technical support and Leny Dimitrova for support with the collection of data.

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
