## [Reviewer comments · Royal Society Open Science]

Review History

RSOS-201100.R0 (Original submission)

Review form: Reviewer 1

Is the manuscript scientifically sound in its present form?

Yes

Are the interpretations and conclusions justified by the results?

Yes

Is the language acceptable?

Yes

Do you have any ethical concerns with this paper?

No

Have you any concerns about statistical analyses in this paper?

Yes

Recommendation?

Accept with minor revision (please list in comments)

Comments to the Author(s)

The authors are interested in the effects of real-world image content (urban vs natural scenes) on cognitive load and walking behaviour. This well-conducted study shows that urban scenes are detrimental to these measures. This result is of general interest and specific interest to people working on gait, environment design and the statistics of real-world images. I have just a few points for the authors to consider as follows:

1. One of the difficulties in studies such as these is how can you be sure that the independent variable you have manipulated (natural vs urban scenes) is the critical factor for any differences observed in the results. The authors are wary of this problem and balance their image sets for path content and distance content. I would like to know a bit more about how these decisions were made. Were the images just rated by the experimenters? How was very close, close, medium, far and very far calibrated, and can we be sure that these distances will have the same meaning across the natural and urban images?
2. P6. I did wonder whether a figure/schematic of the experimental set up /lab might help the reader. I don't think it is essential, but it would have helped me orient myself with the task more readily.
3. Exclusion trials. P7 L1. We are told that one participant was removed because they did not follow instructions. Then on L7/8 we are told this again. Perhaps this second reference is only about individual trials – e.g. some participants messed up occasionally on the odd few trials. Were these trials repeated? This could be clearer.
4. Also (similar location to above) 'computer errors' made me a bit nervous. Perhaps this can be unpacked a little – I'm guessing these were occasional problems with motion tracking and not programming errors?
5. Fig 2. The main interest here is a comparison across urban and nature. There is a significant effect for both velocity and step length, though the effects are quite small. It would help this reader to judge the magnitude of this effect if I knew what the SD of these measures was across the group and also the average SD across trials (of the same condition) within participant.

Review form: Reviewer 2

Is the manuscript scientifically sound in its present form?

Yes

Are the interpretations and conclusions justified by the results?

Yes

Is the language acceptable?

Yes

Do you have any ethical concerns with this paper?

No

Have you any concerns about statistical analyses in this paper?

Yes

Recommendation?

Accept as is

Comments to the Author(s)

Great study, very comprehensive. I would like you to consider the colour contrast distributions in your images and discomfort but that is for the future, not now. It's lengthy, but well expressed. S-cone contribution to discomfort? Worth looking at if you have the data.

Decision letter (RSOS-201100.R0)

Dear Miss Burtan

On behalf of the Editors, we are pleased to inform you that your Manuscript RSOS-201100 "The nature effect in motion: visual exposure to environmental scenes impacts cognitive load and human gait kinematics" has been accepted for publication in Royal Society Open Science subject to minor revision in accordance with the referees' reports. Please find the referees' comments along with any feedback from the Editors below my signature.

Please submit your revised manuscript and required files (see below) no later than 7 days from today's (ie 12-Nov-2020) date. Note: the ScholarOne system will 'lock' if submission of the revision is attempted 7 or more days after the deadline. If you do not think you will be able to meet this deadline please contact the editorial office immediately.

on behalf of Dr Isabelle Mareschal (Associate Editor) and Essi Viding (Subject Editor)
openscience@royalsociety.org

Associate Editor Comments to Author (Dr Isabelle Mareschal):

Two expert reviewers have provided feedback on your manuscript. Please provide a point reply to their queries, including a clarification of the 3Minkowski-Bouligand fractal dimension box-counting technique in the manuscript.

Reviewer comments to Author:

Reviewer: 1

Comments to the Author(s)

The authors are interested in the effects of real-world image content (urban vs natural scenes) on cognitive load and walking behaviour. This well-conducted study shows that urban scenes are detrimental to these measures. This result is of general interest and specific interest to people working on gait, environment design and the statistics of real-world images. I have just a few points for the authors to consider as follows:

1. One of the difficulties in studies such as these is how can you be sure that the independent variable you have manipulated (natural vs urban scenes) is the critical factor for any differences observed in the results. The authors are wary of this problem and balance their image sets for path content and distance content. I would like to know a bit more about how these decisions were made. Were the images just rated by the experimenters? How was very close, close, medium, far and very far calibrated, and can we be sure that these distances will have the same meaning across the natural and urban images?
2. P6. I did wonder whether a figure/schematic of the experimental set up /lab might help the reader. I don't think it is essential, but it would have helped me orient myself with the task more readily.
3. Exclusion trials. P7 L1. We are told that one participant was removed because they did not follow instructions. Then on L7/8 we are told this again. Perhaps this second reference is only about individual trials - e.g. some participants messed up occasionally on the odd few trials. Were these trials repeated? This could be clearer.
4. Also (similar location to above) 'computer errors' made me a bit nervous. Perhaps this can be unpacked a little - I'm guessing these were occasional problems with motion tracking and not programming errors?
5. Fig 2. The main interest here is a comparison across urban and nature. There is a significant effect for both velocity and step length, though the effects are quite small. It would help this reader to judge the magnitude of this effect if I knew what the SD of these measures was across the group and also the average SD across trials (of the same condition) within participant.

Reviewer: 2

Comments to the Author(s)

Great study, very comprehensive. I would like you to consider the colour contrast distributions in your images and discomfort but that is for the future, not now. It's lengthy, but well expressed. S-cone contribution to discomfort? Worth looking at if you have the data.

===PREPARING YOUR MANUSCRIPT===

===PREPARING YOUR REVISION IN SCHOLARONE===

-- Ensure that your data access statement meets the requirements at <https://royalsociety.org/journals/authors/author-guidelines/#data>. You should ensure that you cite the dataset in your reference list. If you have deposited data etc in the Dryad repository, please only include the 'For publication' link at this stage. You should remove the 'For review' link.

Author's Response to Decision Letter for (RSOS-201100.R0)

See Appendix A.

Decision letter (RSOS-201100.R1)

Dear Miss Burtan,

It is a pleasure to accept your manuscript entitled "The nature effect in motion: visual exposure to environmental scenes impacts cognitive load and human gait kinematics" in its current form for publication in Royal Society Open Science. The comments of the reviewer(s) who reviewed your manuscript are included at the foot of this letter.

on behalf of Dr Isabelle Mareschal (Associate Editor) and Essi Viding (Subject Editor)
openscience@royalsociety.org

Appendix A

School of Psychological Science

12a Priory Road
Bristol BS8 1TU
U.K.

☎ : +44 117 9288571
Fax : +44 117 9288588
daria.burtan@bristol.ac.uk

Daria Burtan
PhD candidate

Bristol, 19/11/2020

RE: Revision of manuscript RSOS-201100

Dear Dr Mareschal (dear Isabelle),

We would like to thank you and our reviewers for their interest in our manuscript and the thoughtful feedback which we have incorporated in the revised version of our manuscript (Manuscript RSOS-201100). Please find the revision attached for consideration for publication in *Royal Society Open Science*.

Below is a point-by-point response to the issues raised by yourself and the reviewers. We hope that our revisions make the manuscript now acceptable for publication.

Warmest regards

Daria Burtan and Ute Leonards

Reply to Reviewers:

Associate Editor Comments to Author (Dr Isabelle Mareschal):

Two expert reviewers have provided feedback on your manuscript. Please provide a point reply to their queries, including a clarification of the Minkowski–Bouligand fractal dimension box-counting technique in the manuscript.

In the revised manuscript, we added a short explanation of the procedure used for our box-counting technique “: *after normalising colour images and converting them into grayscale images, images were binarized using the mean image value, before running a box counting algorithm over a range of box sizes to calculate fractal dimensions.*” (page 5 / lines 5-8)

Reviewer 1:

The authors are interested in the effects of real-world image content (urban vs natural scenes) on cognitive load and walking behaviour. This well-conducted study shows that urban scenes are detrimental to these measures. This result is of general interest and specific interest to people working on gait, environment design and the statistics of real-world images. I have just a few points for the authors to consider as follows :

1. One of the difficulties in studies such as these is how can you be sure that the independent variable you have manipulated (natural vs urban scenes) is the critical factor for any differences observed in the results. The authors are wary of this problem and balance their image sets for path content and distance content. I would like to know a bit more about how these decisions were made. Were the images just rated by the experimenters? How was very close, close, medium, far and very far calibrated, and can we be sure that these distances will have the same meaning across the natural and urban images?

We fully agree with the reviewer that image selection is a tricky point in a study such as this. Images were selected from a far bigger image set, allowing us to visually match images for their layout by taking the presence of a walkable path, distance and overall image configuration into account. Selection was based on agreement between three of the experimenters. We believe that such perceptual judgements make it far more likely that the meaning of distance across natural and urban images is preserved than any objective distance measures. In the revised version of the manuscript, we made this perceptual (and thus subjective) image selection even more explicit (page 4 / lines 18 to 30).

2. P6. I did wonder whether a figure/schematic of the experimental set up/lab might help the reader. I don't think it is essential, but it would have helped me orient myself with the task more readily. We discussed the addition of a schematic between authors but felt that it would not sufficiently improve the readability of the manuscript to warrant inclusion.

3. Exclusion trials. P7 L1. We are told that one participant was removed because they did not follow instructions. Then on L7/8 we are told this again. Perhaps this second reference is only about individual trials – e.g. some participants messed up occasionally on the odd few trials. Were these trials repeated? This could be clearer. We thank the reviewer for making us aware of this lack of clarity. Indeed, the second reference referred to the occasional trial of individuals (e.g. participants did not walk all the way to the end of the laboratory but stopped before). Such trials were not repeated as data loss became only apparent at analysis stage. This section has now been rephrased (page 7/ lines 11-16).

4. Also (similar location to above) 'computer errors' made me a bit nervous. Perhaps this can be unpacked a little – I'm guessing these were occasional problems with motion tracking and not programming errors?

These errors were due to occasional synchronisation issues between stimulus projection computer and the motion capture system. This has now been clarified in the manuscript. We also added that affected trials were repeated. (page 7/ lines 16-17)

5. Fig 2. The main interest here is a comparison across urban and nature. There is a significant effect for both velocity and step length, though the effects are quite small. It would help this reader to judge the magnitude of this effect if I knew what the SD of these measures was across the group and also the average SD across trials (of the same condition) within participant.

As with all gait measures, variability of both velocity and step length within the same condition is larger across participants due to differences in leg length and preferred walking speed than it is within an individual. For example, for the neutral condition (i.e. walking toward gray background images), individual SD ranged between 1.3 and 7cm/s for velocity (average individual SD of 4.5cm/s) and between 0.6 and 4cm for step length (average individual SD of 1.3cm). Group SD across conditions were 8.6cm/s for velocity and 3.7 cm for step length. As we have our own control condition included and present not only the ANOVA results but also multilevel modelling that takes into account intra- and inter-individual variability as random effects, we did not feel that an additional table with standard deviations would add to the manuscript.

Note, however, that the observed velocity of 1.34m/s (± 0.065 m/s SD) for the neutral condition corresponds well with the normal walking speed of 1.33m/s (± 0.17 m/s SD) on flat ground reported in the literature (see e.g. table 2 in Waters et al., 1988, Energy-Speed Relationship of Walking: Standard Tables; *J.Orthopaed.Res.*, 6, 215-222). Walking toward both nature (1.3m/s ± 0.063 m/s SD) and urban scenes (1.28m/s ± 0.065 m/s SD) was thus around 10% slower. Also, the average step length of 0.69m (± 0.065 m SD) observed in our neutral condition is well in line with reports of an average step length of 0.71m (± 0.084 m SD) for normal walking (see table 2 in Waters et al., 1988). Shorter steps for walking toward nature and urban scenes are seen as is in line with decreased walking speed.

Reviewer 2

Great study, very comprehensive. I would like you to consider the colour contrast distributions in your images and discomfort but that is for the future, not now. It's lengthy, but well expressed. S-cone contribution to discomfort? Worth looking at if you have the data.

We are delighted that the reviewer is as enthusiastic about this study as we are and would like to thank them for their suggestions. We haven't looked at S-cone contribution and its relation to discomfort in this particular data set (as comparable studies use far higher image numbers to do so). However, a different study in our lab (not yet submitted) on the amount of chlorophyll / greenery in abstract images did neither show any colour contrast distribution-specific effects on gait nor on visual discomfort.